# New Phosphorous-Based [FeFe]-Hydrogenase Models

**Florian Wittkamp [1], Esma Birsen Boydas [2], Michael Roemelt [2] and Ulf-Peter Apfel [1,3,*]**

1   Lehrstuhl für Anorganische Chemie I, Ruhr-Universität Bochum, Universitätsstraße 150, 44801 Bochum, Germany; florian.wittkamp@rub.de
2   Lehrstuhl für Theoretische Chemie, Ruhr-Universität Bochum, Universitätsstraße 150, 44801 Bochum, Germany; esma.boydas@theochem.ruhr-uni-bochum.de (E.B.B.); michael.roemelt@theochem.rub.de (M.R.)
3   Department of Electrosynthesis, Fraunhofer UMSICHT, Osterfelder Str. 3, 46047 Oberhausen, Germany
*   Correspondence: ulf.apfel@rub.de; Tel.: +49-(0)234-32-21831

**Abstract:** [FeFe]-hydrogenases have attracted research for more than twenty years as paragons for the design of new catalysts for the hydrogen evolution reaction (HER). The bridging dithiolate comprising a secondary amine as bridgehead is the key element for the reactivity of native [FeFe]-hydrogenases and was therefore the midpoint of hundreds of biomimetic hydrogenase models. However, within those mimics, phosphorous is barely seen as a central element in the azadithiolato bridge despite being the direct heavier homologue of nitrogen. We herein synthesized three new phosphorous based [FeFe]-hydrogenase models by reacting dithiols $(HSCH_2)_2P(O)R$ (R = Me, OEt, OPh) with $Fe_3(CO)_{12}$. All synthesized mimics show catalytic reactivity regarding HER and change their mechanisms depending on the strength of the used acid. In all presented mimics, the oxide is the center of reactivity, independent of the nature of the bridgehead. However, the phosphorous atom might be reduced by the methods we present herein to alter the reactivity of the model compounds towards protons and oxygen.

**Keywords:** [FeFe]-hydrogenase mimic; element homologue replacement; reduction of phosphinates and phosphines; cyclic voltammetry; Mössbauer spectroscopy; infrared spectroscopy

## 1. Introduction

[FeFe]-hydrogenases and model compounds thereof have been extensively studied over the past few years and are also investigated in regards to element homologue replacement (EHR). The active site of [FeFe]-hydrogenases, the so-called "H-cluster", consists of a cubane-like [4Fe-4S] cluster and a [2Fe-2S] cluster, which bears additional CO and CN ligands and is bridged by an azadithiolate (adt) (Figure 1a) [1–4]. Since the groups of Happe, Lubitz and Fontecave found that [2Fe]$_H$ can be replaced by synthetic cofactors, much effort has been made to synthesize a variety of different cofactors that can be placed into the enzyme [5–7]. However, only mimics that comprise an amine as the bridgehead, i.e., ADT (native activity), ADSe (100%) and a monocyanide of ADT (50%), result in an active enzyme after maturation [7,8]. Besides variation of the bridging dithiol to pdt, odt, edt and sdt, there are two more models that pick up the principle EHR. One example is Ru-ADT, in which both iron atoms are exchanged to ruthenium [9]. Although being inactive for $H_2$ oxidation and evolution, this variant has the advantage to trap the Hhyd state. The other example is the formerly mentioned ADSe, in which the dithiolate bridge is replaced by a diselenolate bridge [8]. This hybrid enzyme is active for proton reduction and $H_2$ oxidation but is more biased towards the former. The chalcogenide exchange further results in a dramatically reduced oxygen stability and, consequently, an even more sensitive H-cluster than the native cofactor.

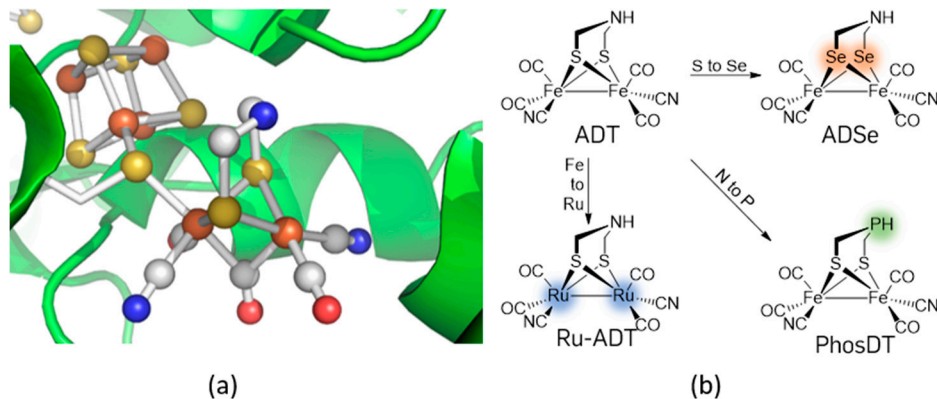

**Figure 1.** (**a**) Active site of [FeFe]-hydrogenases known as H-Cluster. Color code: white: carbon, blue: nitrogen, red: oxygen, yellow: sulfur, orange: iron; (**b**) Alterations within the diiron subsite based on the principle of element homologue replacement. Therein, the ADSe and Ru-ADT derivatives are already known from our and other groups.

The third moiety that can be altered by an EHR is the secondary amine of the bridging dithiolate. A phosphine bridgehead would be the next heavier homologue of this amine and most likely result in a model compound with changed oxygen tolerance, reactivity/state accessibility and proton transport activity. Nitrogen and phosphorus show very different chemical behavior, although both are non-metal elements from the fifth main row. In particular, the high oxygen affinity of phosphorous compared to the inertness of nitrogen describes these differences perfectly. While more than 99% of the worldwide nitrogen abundance stems from inert $N_2$ (oxidation state ±0), phosphorous occurs natively in derivatives of $H_3PO_4$, in which P is commonly in the oxidation state +V (Figure 2), illustrating the different reactivity. However, phosphorus in its oxidation state +III can replace nitrogen in organic compounds, showing their similar structural features in this oxidation state as well (Table 1). However, although they are structurally equivalent to their lighter homologues, organo phosphorous compounds ($P^{III}$) show a different reactivity. These phosphines tend to readily oxidize to form the more stable phosphine oxides, [10] whereas amines are hard to oxidize and such an oxidation requires the use of strong oxidants, e.g., $H_2O_2$ [11]. Furthermore, phosphorous compounds with double- and triple bonds between P and C show a reactivity more comparable to alkenes and alkynes than to imines and nitriles due to the smaller difference in electronegativity ($\Delta E_N$) of phosphorous (2.19) and carbon (2.55) compared to nitrogen (3.04) and carbon [12]. The small $\Delta E_N^{C-P}$ is also responsible for the aromatic character of phosphinines, although they are air- and moisture sensitive, which is not the case for pyridine. Opposite to this, phospholes do not show any aromaticity, since the lone pair of phosphorous is not delocalized, which is not the case for pyrroles that show aromatic behavior. These examples show how element homologue replacement (EHR), i.e., N to P, can affect the reactivity, oxygen sensitivity and electronic structure of a molecular system.

$$
\begin{array}{cccc}
\overset{O}{\underset{R''O}{R'O-\overset{\|}{P}-OR'''}} & \overset{O}{\underset{R''O}{R'O-\overset{\|}{P}-R'''}} & \overset{O}{\underset{R''}{R'O-\overset{\|}{P}-R'''}} & \overset{O}{\underset{R''}{R'-\overset{\|}{P}-R'''}} \\
\text{Phosphates} & \text{Phosphonates} & \text{Phosphinates} & \text{Phosphine oxides}
\end{array}
$$

+V

$$
\begin{array}{cccc}
\underset{R''O}{R'O-\overset{|}{P}-OR'''} & \underset{R''O}{R'O-\overset{|}{P}-R'''} & \underset{R''}{R'O-\overset{|}{P}-R'''} & \underset{R''}{R'-\overset{|}{P}-R'''} \\
\text{Phosphites} & \text{Phosphonites} & \text{Phosphinites} & \text{Phosphines}
\end{array}
$$

+III

**Figure 2.** Overview of organo phosphorus compounds in oxidation state +V as well as their reduced analogues.

**Table 1.** Overview of some organo phosphorous (P$^{+III}$) compounds and structural related amine compounds.

|  | P-Type | P—Compounds | N-Type | N—Compounds |
|---|---|---|---|---|
| Aliphatics | Phosphines | $PR_3$ | Amines | $NR_3$ |
|  | Phosphaalkenes | $R_2C{=}PR$ | Imines | $R_2C{=}NR$ |
|  | Phosphaalkynes | $R_2C{\equiv}P$ | Nitriles | $R_2C{\equiv}N$ |
|  | Diphosphenes | RP=PR | Azo compounds | RN=NR |
| Aromatics/ Heterocycles | Phosphinines |  | Pyridines |  |
|  | Phospholes |  | Pyrroles |  |

However, there is only one model known that comprises a phosphorous bridge. In 2010, Song and coworkers attempted to synthesize $[Fe_2(CO)_6][(\mu\text{-}SCH_2)_2PR]$ (R = Ph, $\eta^5$-C$_5$H$_4$CH$_2$FeCp) using the tertiary phosphine $RP(CH_2OH)_2$ and $(\mu\text{-}HS)_2Fe_2(CO)_6$ but found that the phosphine replaces one CO ligand to afford **A** ($Fe_2S_2(CO)_5PR_3$, Figure 3) due to the high nucleophilicity of the phosphine ligand [13].

**Figure 3.** Phosphorous containing models of the diiron subsite of [FeFe]-hydrogenases [13,14].

Subsequently, Weigand and coworkers took up this result and synthesized $PhP(O)(CH_2Cl)_2$ as the possible bridge and received **B** ($[Fe_2(CO)_6][(\mu\text{-}SCH_2)_2P(O)Ph]$, Figure 3) after reacting it with $(\mu\text{-}LiS)_2Fe_2(CO)_6$ [14]. This complex is the hitherto only described [FeFe]-hydrogenase model comprising any phosphorous atom as the bridgehead atom. However, due to the bulky P(O)Ph moiety and the hydrophobic phenyl substituent, **B** unlikely fits into the binding pocket of the [FeFe]-hydrogenase, which renders this model compound unsuitable for functional hybrid enzyme modifications. Furthermore, the basicity of the bridgehead, which is a prerequisite for an efficient proton transfer from the proton transfer path inside the protein to the iron center, where the proton is reduced to a hydride, was found to be a key aspect for a fully active enzyme.

The native cofactor ADT exhibits a pK$_a$ of approximately eight [15,16], which is the pK$_a$ one should aim for in the model compounds. The mimic presented by Weigand and coworkers, however, must be protonated by HBF$_4 \cdot$OEt$_2$, a very strong acid, indicating a low pK$_a$ value of the P(O)Ph moiety [14]. We herein present a set of novel [FeFe]-hydrogenase model compounds that comprise phosphine oxide, phosphinic acid and phosphinates within the bridge. In addition, we herein describe our attempts to form the hitherto unprecedented complexes D–F (Figure 4). Notably, following literature reported basicities for various phosphine derivatives, these complexes possess a broad range of pK$_a$ values (Figure 4). Thus, we anticipate that the suggested model compounds are interesting replacements for the native cofactor regarding stability and reactivity.

**Figure 4.** [FeFe]-hydrogenase model compounds comprising phosphadithiol bridgeheads D–F and the hexacarbonyl precursor of the native cofactor ADT, C. The known/expected pKa values are noted under each structure. $pK_a$ values for **C** are from ref. [16]. Approximated values for D–F are based on dimethylphosphine (**D**) [17], trimethylphosphine (**E**) [17] and dialkylphosphinic acids (**F**).

## 2. Results

### 2.1. Synthesis of Phosphinates and Phosphine Oxides as Well as the Iron Carbonyl Complexes Thereof

Scheme 1 presents the various reaction steps for the phosphine oxides **1–10**. Step A is a condensation reaction between phosphinic acid and paraformaldehyde. Concerning the following chlorination of the hydroxy groups, yielding **2**, Maier and coworkers already noted that the anhydride is formed upon reacting **1** at room temperature. Upon heating the reaction mixture to reflux, the anhydride can be converted to the chloride **2** [18]. Therefore, **1** was directly added very slowly to refluxing thionyl chloride. Step C is a hydrolyzation reaction of an acid chloride that was performed by a slow addition of **2** to distilled water. The formation of compounds **3-R** (step E) is an esterification that was catalyzed by NEt₃. After NEt₃·HCl was filtered off and the solvent removed, compounds **3-R** were purified to afford colorless oils in good yield and high purity. Besides the phosphinic acid derivative **5** and its esters **3-R**, methyl phosphine **4** was prepared via Grignard reaction with MeMgBr (step H).

**Scheme 1.** Pathway for the synthesis of phosphorous containing dithiol ligands starting from phosphinic acid. Conditions of single steps: (**A**) 6 h 40 °C, 30 h reflux; (**B**) 6 h reflux; (**C**) 5 min RT; (**D**) (i) 6 h 110 C, (ii) 1 h 100 °C; (**E**) Additions at 0 °C, 4 h RT; (**F**) 16 h RT; (**G**) 0.5 h RT, neutralized with HCl; (**H**) Addition at 0 °C, 4 h reflux, quenched with sat. NaCO₃; (**I**) 16 h RT; (**J**) 0.5 h RT, neutralized with HCl.

We found that the phosphinate **3-Et** is especially easily accessible and reacted it with Li₂[Fe₂S₂(CO)₆] to afford [Fe₂(CO)₆][(μ-SCH₂)₂P(O)OEt] (**12-OEt**) (Scheme 2, Method A). The purification of the [FeFe]-hydrogenase model compound was performed by column chromatography with SiO₂ as stationary

phase and a 1:1 mixture of PE/EtOAc as the mobile phase to get a red crystalline solid in 5% yield after removing the solvent.

**Scheme 2.** Principle methods for synthesizing [FeFe]-hydrogenase models starting from $Fe_2S_2(CO)_6$ (Method A) or from $Fe_3(CO)_{12}$ (Method B).

The IR spectrum of **12-OEt** shows four sharp CO bands at 2079, 2044, 2004 and 1971 $cm^{-1}$, which are within ±10 $cm^{-1}$ discrepancy in the same range as those observed for $[Fe_2(CO)_6][(\mu-SCH_2)_2NH]$, the precursor of the native cofactor, showing a comparable electron density at the iron centers.

Even though we could effectively obtain a [FeFe]-hydrogenase model **12-OEt** with a phosphinate bridgehead using method A, it is not the most efficient way due to low yields and the required laborious synthesis and workup of $Fe_2S_2(CO)_6$ (**11**) [19]. We therefore aimed for the synthesis of bis(mercaptomethyl)phosphine oxides and -phosphinates instead of their bis(chloromethyl) derivatives, which would produce the same models using $Fe_3(CO)_{12}$, a commercially available iron carbonyl, as the starting material (Scheme 2, Method B).

In the case of **3-Et** and **3-Ph**, nucleophilic substitution with potassium thioacetate was found to be the suitable method to introduce a protected thiol. Thus, the chlorides were dissolved in DMF and 2.5 equivalents of KSAc were added as solid. The mixture was stirred for 16 h at room temperature to afford a dark red solution that was condensed to a greasy solid. Solvation in water and extraction with DCM gave the respective thioacetates **7-Et** and **7-Ph** with 78% and 86% yield, respectively. For the phosphinic acid **6**, the addition of thiourea was used instead. Here, chloride **5** was dissolved in n-butanol and two equivalents of thiourea were added as solid. While stirring the mixture at 110 °C, the product forms as a white solid and was filtered off and washed with fresh n-butanol after 6 h. In all cases, deprotection with sodium hydroxide and following acidification of the reaction mixture gave the free thiols **6**, **8** and **10**, which were then purified via bulb-to-bulb distillation and used to synthesize the [FeFe]-hydrogenase models according to method B (Scheme 2). The two additional steps to synthesize dithiols **6**, **8** and **10** starting from chlorides **3–5** are far more convenient regarding the reaction protocol than the synthesis of **11** from $Fe(CO)_5$ [19]. Furthermore, method B poses an easy to handle synthesis for [FeFe]-hydrogenase models without the need of the additional reaction step to generate $Li_2[Fe_2S_2(CO)_6]$ from **11** and $LiBHEt_3$, which is very air sensitive and can undergo unwanted side reactions. For example, we observed the formation of $[Fe_4(SEt)_2(S_2)(CO)_{12}]$ upon reacting **11** with $LiBHEt_3$ and a nitro-derivative of **3-Ph**. Therein, the ethyl groups of the reductant are the only ones present in this reaction mixture and thus have to be the source of the SEt moieties. The formation of this dimer was previously observed by Cheng and coworkers, who aimed to obtain the synthesis of an ADT derivative starting from **11** and $LiBHEt_3$ as well [20].

In the case of methyl phosphinate, we found that both the addition of potassium thioacetate or thiourea leads to an unidentified, water-soluble product after workup. Whether this compound is the phosphinic acid derivative **6**, which is water soluble, or a species resulting from decomposition could not be concluded.

Following method B, compounds **12-R** were synthesized by reacting dithiol **6**, **8** and **10** with $Fe_3(CO)_{12}$ in THF at room temperature for 2 h. The purification of **12-OEt** was performed as described for method A, affording the desired compound with 46% yield. However, although the same method was used to synthesize model compound **12-OPh**, it readily undergoes transesterifications with stronger nucleophiles. Thus, eluting with PE/EtOAc produces **12-OEt**, as indicated by NMR, while

elution with DCM/MeOH produces **12-OMe** in very low yields (Figure 5b). Interestingly, the lost phenyl group co-crystallizes with the methyl phosphinate-bridged compound in close proximity (2.7 Å). When changing the mobile phase to DCM/acetone (1:1) to avoid the use of methanol, the same result as for DCM/MeOH was obtained as verified by single crystal XRD. Since $Fe_3(CO)_{12}$ is stabilized with 5%–10% methanol, it is a reasonable assumption that, besides the solvent, the stabilizer can undergo the transesterification as well, which explains the overall low yields for this reaction. The model compound **12-Me**, bearing a phosphine oxide instead of a phosphinate bridgehead, was purified by column chromatography as well. Elution with DCM/MeOH (50:1) gave a red solid with 46% yield after complete removal of the solvent. Unfortunately, mimic **12-OH** from the reaction of **8** with $Fe_3(CO)_{12}$ decomposes readily in solution after column chromatography (MeCN/water 4:1), observed by significant amounts of precipitate. The supernatant solution was investigated by IR spectroscopy, showing the typical CO patterns of [FeFe]-hydrogenase mimics (Figure 6). However, due to the rapid decomposition of **12-OH**, this model compound was not further investigated.

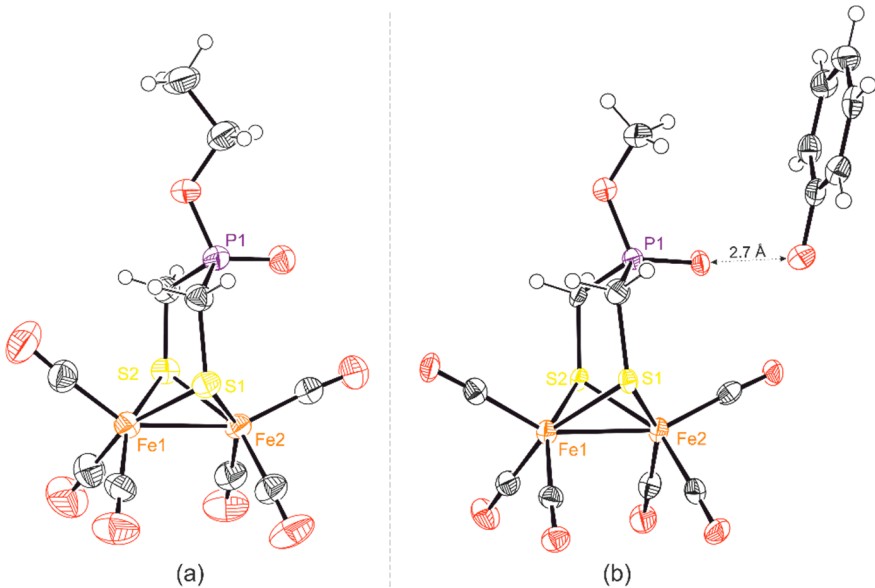

**Figure 5.** (**a**) OPTEP Plot of compound **12-OEt**; (**b**) ORTEP Plot of compound **12-OMe**. Ellipsoids at 50% probability level. **12-OMe** crystallizes in P-1, showing two individual moieties of the iron carbonyl and one phenol. Here, only the phenol-coordinating moiety is shown. For the complete asymmetric unit see Figure S1. Color code: black: carbon, white: hydrogen, red: oxygen, yellow: sulfur, purple: phosphorous.

The molecular structures of **12-OEt** and **12-OMe** are shown in Figure 5 and the structure of phosphine oxide **12-Me** in Figure S2. The $Fe_2S_2$ core of all complexes show the typical butterfly-like structure of [FeFe]-hydrogenase model compounds. The phosphorous atoms of both phosphinates **12-OEt** and **12-OMe** and the phosphine oxide **12-Me** have a slightly distorted tetrahedral geometry and causes average P-C-S bond angles of 117.02° and 115.6° for **12-OEt** and **12-OMe** respectively. The phosphine oxide **12-Me** exhibits a respective angle of 115.9°. The P=O distance of all compounds is 1.45 ± 0.03 Å. Taking these values into account, no structural difference between model compounds **12-OEt** and **12-OMe** can be observed, both comprising a phosphinate bridge and **12-Me**, bearing a methylphosphine oxide. Additionally, the shown angle and P=O distance are in line with the previously reported mimic **B**, which bears a phosphine oxide bridge as well [14].

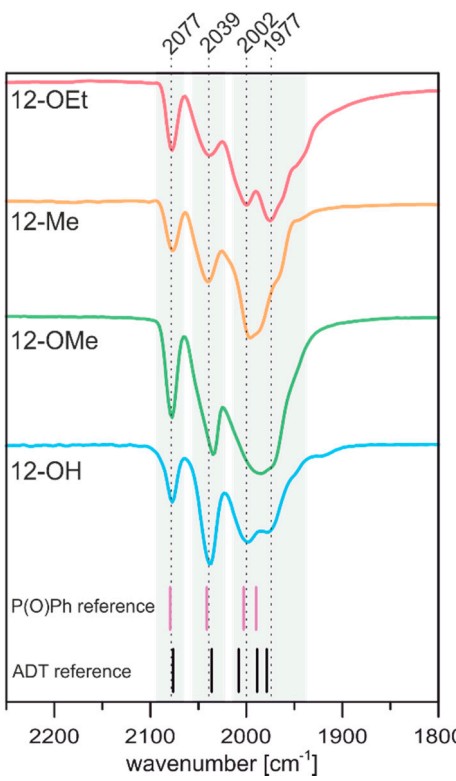

**Figure 6.** ATR-IR spectra of compounds **12-R** in comparison to model compounds B of Weigand and coworkers (P(O)Ph reference [14]) and the native cofactor's precursor $[Fe_2(CO)_6][(\mu\text{-}SCH_2)_2NH]$ (ADT, reference [21]).

## 2.2. Spectroscopic Properties of Compounds *12-R*

### 2.2.1. Infrared Spectroscopy

The CO stretching bands of [FeFe]-hydrogenase mimics are of utmost interest, showing differences in electron density and ligand orientation. Figure 6 presents the respective spectra of the oxidized compounds **12-R**. All four compounds show two sharp main bands at 2077 and 2039 $cm^{-1}$, as well as a broader contribution around 1990 $cm^{-1}$ that is more or less dissolved. The overall band positions are in line with the native cofactor mimic **C** and the phenylphosphine oxide model **B**. These models show IR patterns at 2076, 2036, 2008 and 1989 and 1979 $cm^{-1}$ and at 2079, 2041, 2003 and 1990 $cm^{-1}$, respectively (Figure 6) [14,21]. Due to the similar CO band positions, it can be thought that all P-containing models and ADT exhibit a comparable electron density at the iron atoms. Therefore, the exchange of nitrogen to its heavier homologue, phosphorous, seems to not affect the electronics of the iron atoms at a first glance.

### 2.2.2. Mössbauer Spectroscopy

A second tool to investigate the electronics of [FeFe]-hydrogenase models is Mössbauer spectroscopy. While the isomer shift $\delta$ is proportional to the electron density at the respective nucleus, iron in this case, the asymmetric shape of the electronic charge around the nucleus and the nucleus itself determines the quadrupole splitting $\Delta E_Q$. The former is expressed by the electronic field gradient (EFG). For $[Fe_2(CO)_6][(\mu\text{-}SCH_2)_2NH]$, these parameters are $\delta = 0.05$ mm/s and $\Delta E_Q = 0.64$ mm/s, respectively, indicating a low-spin Fe(I) system (Table 2, entry 7). All [FeFe]-hydrogenase models with altered dithiol bridges do resemble these parameters relatively well, particularly in the case of all-CO mimics, indicating unchanged electronic properties of the iron atoms. (Table 2, entries 4–8). Upon changing the carbonyl ligands to phosphines, however, $\delta$ and $\Delta E_Q$ substantially change compared to all-CO and dicyanide

models, as can be observed for [Fe$_2$(μCO)(CO)$_2$(PMe)(dppe)][(μ-SCH$_2$)$_2$)]. In this case, two different isomer shifts indicate a different electron density at each of the two iron atoms, which is a result of unsymmetrical substitution of the metals (Entry 16) [22].

**Table 2.** Mössbauer parameter of compounds **12-R** compared to different known model complexes.

| Entry | Compound | δ [mm/s] | ΔE$_Q$ [mm/s] | Ref. |
|:-----:|:---------|:--------:|:-------------:|:----:|
| 1 | [Fe$_2$(CO)$_6$][(μ-SCH$_2$)$_2$P(O)OEt] (**12-OEt**) | 0.07 | 0.91 | a) |
| 2 | [Fe$_2$(CO)$_6$][(μ-SCH$_2$)$_2$P(O)OMe] (**12-OMe**) | 0.05 | 0.87 | a) |
| 3 | [Fe$_2$(CO)$_6$][(μ-SCH$_2$)$_2$P(O)Me] (**12-Me**) | 0.05 | 0.87 | a) |
| 4 | [Fe$_2$(CO)$_6$][(μ-SCH$_2$)$_2$CH$_2$] (PDT) | 0.05 | 0.63 | a) |
| 5 | [Fe$_2$(CO)$_6$][(μ-SeCH$_2$)$_2$CH$_2$] (PDSe) | 0.06 | 0.65 | a) |
| 6 | [Fe$_2$(CO)$_6$][(μ-SeCH$_2$)$_2$NH] (ADSe) | 0.07 | 0.65 | a) |
| 7 | [Fe$_2$(CO)$_6$][(μ-SCH$_2$)$_2$NH] (ADT) | 0.05 | 0.64 | a) |
| 8 | [Fe$_2$(CO)$_6$][(μ-SCH$_2$)$_2$Ph] (BDT) | 0.08 | 0.46 | a) |
| 9 | [Fe$_2$(CO)$_4$(CN)$_2$][(μ-SCH$_2$)$_2$CH$_2$]$^{2-}$ | 0.06 | 0.92 | a) |
| 10 | [Fe$_2$(CO)$_4$(CN)$_2$][(μ-SCH$_2$)$_2$NH]$^{2-}$ | 0.09 | 0.43 | a) |
| 11 | [Fe$_2$(CO)$_4$(CN)$_2$][(μ-SeCH$_2$)$_2$CH$_2$]$^{2-}$ | 0.07 | 0.83 | a) |
| 12 | HydA1$^{adt}$ | 0.16, 0.08 | 0.89, 0.55 | [23] |
| 13 | [Fe$_2$(CO)$_6$][(μ-SCH$_2$)$_2$SiMe$_2$] | −0.04 | 0.78 | [24] |
| 14 | [Fe$_2$(CO)$_6$][(μ-SCH$_2$)$_2$SiMe(CH$_2$SH)] | −0.04 | 0.82 | [24] |
| 15 | [Fe$_2$(CO)$_6$][(μ-SCH$_2$)$_2$O)] | −0.02 | 0.81 | [25] |
| 16 | [Fe$_2$(μCO)(CO)$_2$(PMe)(dppe)][(μ-SCH$_2$)$_2$)] | −0.58 (PMe), 0.70 (dppe) | 0.2 (PMe), 0.04 (dppe) | [22] |

a) this work.

The Mössbauer spectra of oxidized compounds **12-R** are presented in Figure 7. All models show isomer shifts of δ = 0.05 ± 0.02 mm/s and quadrupole splittings of ΔE$_Q$ = 0.89 ± 0.02 mm/s and therefore resemble the literature known all-CO mimics closely. In the spectrum of **12-Me**, a second species can be observed, which is a direct product of the degradation of **12-Me**. As concluded from IR experiments as well, Mössbauer spectroscopy substantiates the finding that an exchange of the nitrogen atom to different phosphorous species within the bridging dithiol does not affect the electron density at the iron atoms.

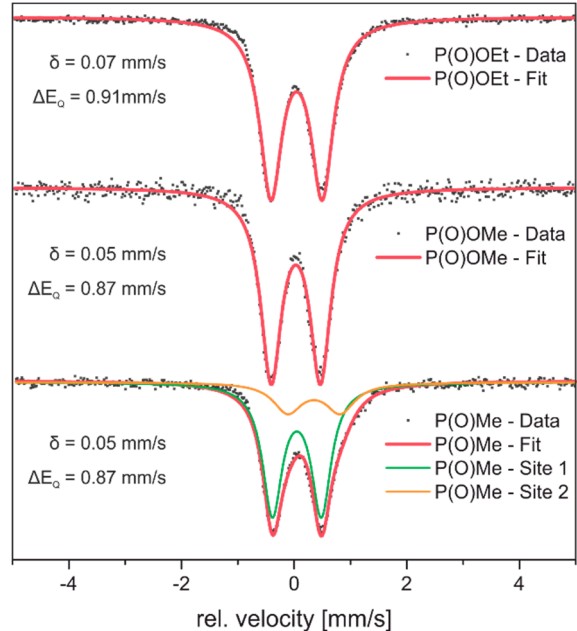

**Figure 7.** Room temperature zero field Mössbauer spectra of compounds **12-OEt**, **12-OMe** and **12-Me**.

*2.3. Reactivity towards Acids*

In acetonitrile, the CO bands of compounds **12-OEt** and **12-Me** slightly upshift to 2081, 2045 and 2006 and 2080, 2044 and 2004 $cm^{-1}$, respectively, compared to the ATR-IR measurements of dried films. These shifts indicate an interaction between the carbonyl ligands and the solvent (Figure 8, blue traces). Since the CO stretching frequency is very sensitive to changes of the electron density of the central atom to which the carbonyls are bound, slight shifts of the resulting bands in the IR spectra indicate, e.g., protonation of the investigated molecules. For [FeFe]-hydrogenase model compounds, the results of such experiments are used to determine potential protonation sites and $pK_a$ values of the compounds.

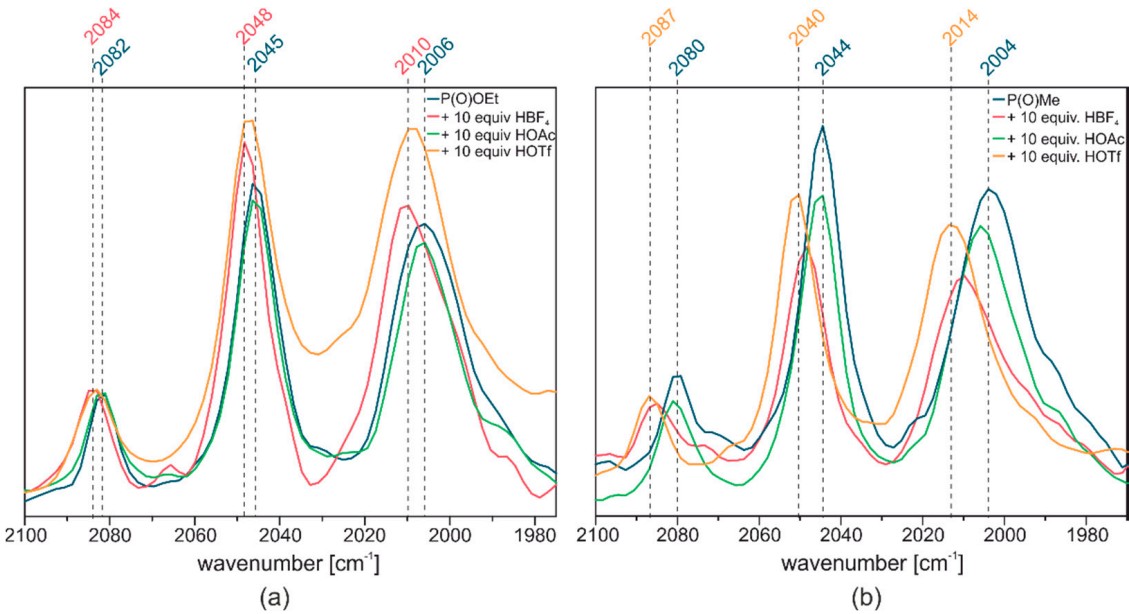

**Figure 8.** (**a**) IR spectrum of $[Fe_2(CO)_6][(\mu\text{-}SCH_2)_2P(O)OEt]$; (**b**) IR spectrum of $[Fe_2(CO)_6]$ $[(\mu\text{-}SCH_2)_2P(O)Me]$. Both compounds were measured as 20 mM solution in acetonitrile in absence (blue) or presence of acids of different strengths (red: $HBF_4$, green: HOAc, yellow: HOTf).

Therefore, acetonitrile solutions of phosphinate **12-OEt** and phosphine oxide **12-Me** were acidified with organic acids of various strength and their resulting IR spectra monitored regarding small shifts. In the case of acetic acid (HOAc), which has a pKa of 23.5 in acetonitrile, [26] no shift of the CO bands can be observed indicating no protonation of the cluster. The use of 10 equivalents of triflic acid (HOTf, pKa = 2.6 [27]) or aqueous $HBF_4$ (= $H_3O^+BF_4^-$, $pK_a$ in MeCN = 2.3 ± 0.1 [28]) however, causes a slight up-shift of all CO bands (~4 $cm^{-1}$), resulting in a new CO pattern of 2084, 2048, 2010 $cm^{-1}$ in the case of **12-OEt**, that does not further change upon adding more acid to the solution (Figure S3). This shift is even more pronounced in the case of **12-Me** (~7 $cm^{-1}$), showing a new pattern of the CO bands at 2087, 2040 and 2014 $cm^{-1}$ after addition of HOTf. Interestingly, the use of aqueous $HBF_4$, although being the slightly stronger acid, results in a less pronounced shift of the CO bands.

Different protonation scenarios were already discussed in literature: While direct metal protonation, yielding a μH binding mode, results in an CO band up shift of up to 100 $cm^{-1}$, [29,30] a smaller shift of approx. +15 $cm^{-1}$ is observed in the case of N-protonation [30,31]. Ligand based protonation, however, e.g., in the case of a phosphaadamantane ligand results in shifts of +18–35 $cm^{-1}$ [32]. While CO protonation was not observed before, P=O protonation was previously observed and calculated by Weigand and coworkers and resulted in shifts of ~10 $cm^{-1}$ [14]. On the basis of the strong structural relation to the model of Weigand, a P=O protonation is suggested for our models as well.

### 2.4. DFT Calculations on the Protonation of [FeFe]-hydrogenase Model Compounds

DFT calculations on **12-OEt** (Figure 9) and **12-Me** (Figure S4) were conducted to probe a possible protonation at the bridging sulfide, the iron center in form of bridging or terminal protons/hydrides and the P=O moiety of the bridgehead. While the former led to more pronounced shifts of the calculated CO frequencies, protonation at the oxide resembles the experimental spectra very well (see SI). Notably, a slight blue shift of all CO bands was observed for the [FeFe]-hydrogenase HydA1 upon acidification under reducing conditions [33,34]. In this case, DFT calculations suggested a protonation at an H-cluster binding cysteine, which was later supported by synchrotron-based techniques [35].

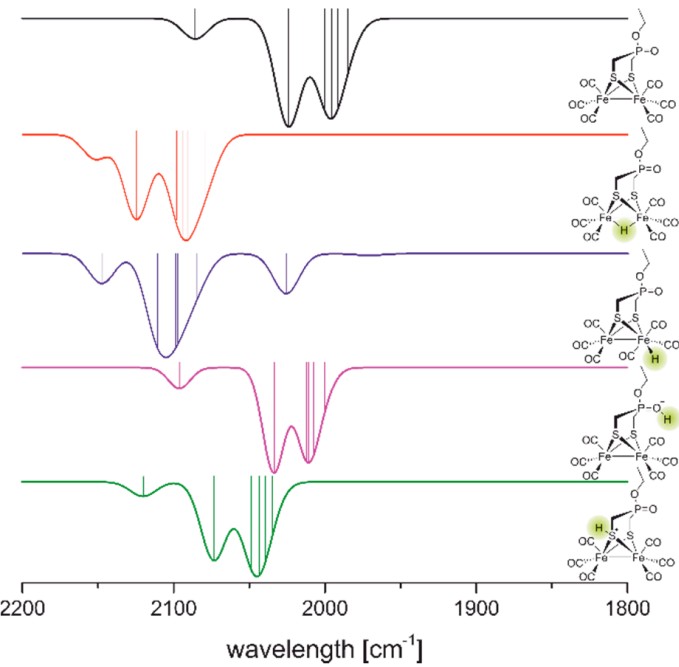

**Figure 9.** DFT derived IR spectra of **12-OEt** in its neutral and protonated form. Protonation sites are highlighted in green for better readability.

### 2.5. Redox Properties of Model Compounds 12-R in Absence and Presence of Acids of Various Strength

In addition to the IR experiments in the presence of different acids, we also performed cyclic voltammetry before and after the addition of acid in which the phosphinate **12-OEt** (Figure 10a), and the phosphine oxide **12-Me** (Figure 10b) behave very similar. The redox behavior of [FeFe]-hydrogenase models is dictated by their first ligand sphere. In the case of the native cofactor's precursor $[Fe_2(CO)_6][(\mu-SCH_2)_2NH]$ the potential for the reduction from a homovalent $Fe^IFe^I$ to a bivalent $Fe^0Fe^I$ complex is at −1.59 V versus the ferrocene/ferrocenium couple. Although bridged by two selenides and down shifted CO bands within the IR spectrum, which suggests higher electron density at the iron centers, ADSe (Figure 1) shows the same reduction potential.

The half peak potential of the [FeFe]-hydrogenase model compounds **12-OEt** and **12-Me** is slightly shifted to more anodic potential, namely −1.42 V and −1.41 V vs. Fc/Fc⁺, respectively. Independent on the nature of the bridgehead, phosphinate **12-OEt** und phosphine oxide **12-Me**, therefore, exhibit the same reduction potential. Figure 10 presents the CVs of **12-OEt** and **12-Me** in presence and absence of acetic acid. As already indicated by IR, both model compounds behave inertly against the weak acid at open circuit potential. Therefore, both complexes, **12-OEt** and **12-Me** have to follow an EC mechanism. After the addition of 10 equivalents of acetic acid, a catalytic reduction wave can be observed at −2.2 V vs. Fc/Fc⁺ after reducing the H₂ase mimic to its $Fe^IFe^0$ state that increases from 10 to 20 equivalents and is at its maximum at 30 equivalents. Notably, **12-OEt** shows a slightly higher current at this potential (−190 μA) than phosphine oxide **12-Me** (−150 μA). The reduction wave does not further increase upon

increasing the amount of acid present in solution. This might indicate a maximum turn-over frequency of both catalysts and "free" protons get reduced at the surface of the glassy carbon electrode. Thus, the observed currents are a result of an overlay of current from a homogeneous and heterogeneous proton reduction and might explain the CV shape between -2.4 V and −2.6 V for the runs with 30+ equivalents of HOAc.

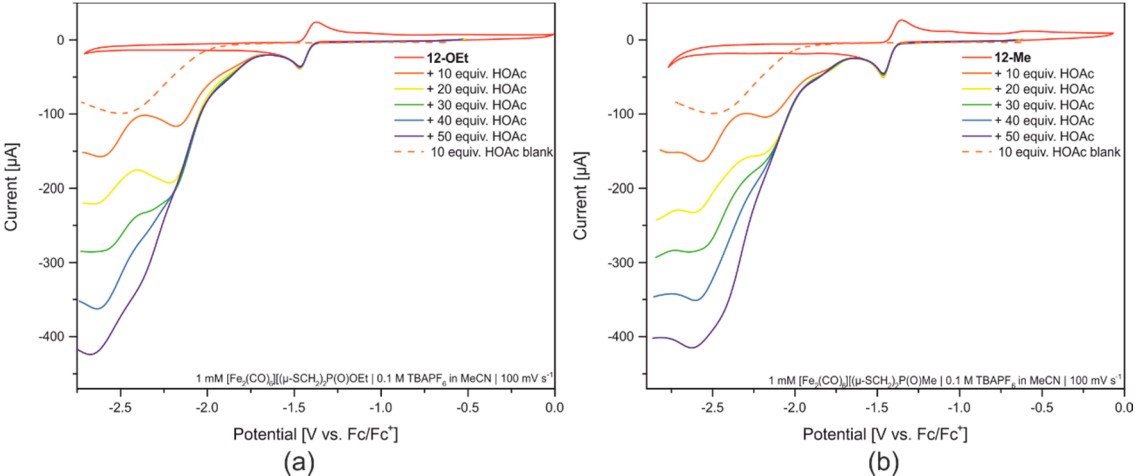

**Figure 10.** (**a**) CVs of **12-OEt**; (**b**) CVs of **12-Me.** Both figures show CVs in presence of 0 to 50 equivalents of acetic acid (HOAc). Both measurements were performed using a three-electrode setup utilizing a glassy carbon as working electrode and platinum-and silver wires as counter- and pseudo-reference electrode, respectively. Both compounds were adjusted to their respective redox potential vs. Fc/Fc$^+$ and used as self-internal reference. Blank measurements were performed using Fc as internal reference.

At approx. −2.6 V vs. Fc/Fc$^+$, the second reduction wave occurs from the reduction to Fe$^0$Fe$^0$ that is not present in acid free medium. However, the protonation of the Fe$^I$Fe$^0$ complex decreases the electron density of the iron and shifts the potential of the second reduction to more positive values.

In the presence of a stronger acid such as H$_3$O$^+$ (see above) however, both compounds change their reactivity, undergoing a CE mechanism (Figure 11 and Figure S5). The potential of the Fe$^I$Fe$^I$/Fe$^I$Fe$^0$ redox couple of both compounds shifts approximately 0.26 V towards more positive potentials upon acidification of the solution, indicating a lower electron density at the iron center evoked by a protonation at the oxide. This result goes in hand with the observed blue shift of the CO bands within the IR spectra upon acidification of **12-OEt** and **12-Me** with H$_3$O$^+$ in acetonitrile. A similar strong shift of +0.2 V was observed for the oxidation of model compound B upon addition of HBF$_4$·OEt$_2$ [14].

Compared with the cyclic voltammograms of **12-OEt** and **12-Me** in presence of HOAc, the catalytic wave is shifted to more positive potentials (−1.5 V vs. Fc/Fc$^+$ for 10 equivalents of acid) after the addition of HBF$_4$ and starts immediately after reduction to Fe$^I$Fe$^0$ (see inlets of Figure 10). Furthermore, the catalytic current of the proton reduction outperforms the respective currents of HOAc experiments by a factor of two, resulting in a current of ~−400 µA for 20 equivalents of acid. The most striking result of the HBF$_4$ experiments is the further increasing catalytic current with increasing amounts of acid. This is not the case for the EC mechanism with the weaker acid HOAc. Thus, the protonation of the P=O moiety does not only shift the reduction potential of the Fe$^I$Fe$^I$/Fe$^I$Fe$^0$ redox couple to more positive values, but seemingly alters the mechanism of catalytic proton reduction in a way that the compounds **12-OEt** and **12-Me** are not turn-over-rate-limited under the tested conditions, and solely diffusion limitations occur in all experiments. To account for the possible two different mechanisms that result from changing the proton source from HOAc to HBF$_4$, we performed spectroelectrochemical measurements on **12-OEt** and combined this experiment with titration experiments of HOAc. The reduction of **12-OEt** can be followed by tracing the CO-band signature shifts within the IR spectrum. The CO-bands shift from 2081, 2046 and 2006 cm$^{-1}$ at open circuit potential to 1975, 1938, 1902 and 1730 cm$^{-1}$ upon applying −1.1 V vs.

Ag/Ag$^+$, which is slightly more negative than the reductive peak potential (Figure S6). The CO-band shift of 100 cm$^{-1}$ is in agreement with the reduction to a Fe$^I$Fe$^0$ complex **[12-OEt]**$^-$ [36]. Interestingly, an IR band at 1730 cm$^{-1}$ is observable upon reducing **12-OEt**. Pickett and coworkers observed a similar behavior of [Fe$_2$(CO)$_6$][(μ-SCH$_2$)$_2$CH$_2$] (PDT) and accounted this signal to a bridging CO [36]. In this case, external CO binds upon the loss of one bridging thiolate, induced by the reduction of PDT and resulting in a dangling HS(CH$_2$)$_3$ chain. In our case, no external CO was added to the analyzed solution of **12-OEt**. Additionally, a substantial reorganization of the CO ligands, which might occur in order to adopt a structure with μCO, seems not to be present according to the identical CO-band structure of the oxidized and reduced **12-OEt**. However, the additional μCO ligand might occur from the degradation of a small amount of substance, resulting in free CO that binds to the intact mimic in a μCO fashion. This "self-cannibalization" process can be regularly found for [FeFe]-hydrogenase enzymes as well [37].

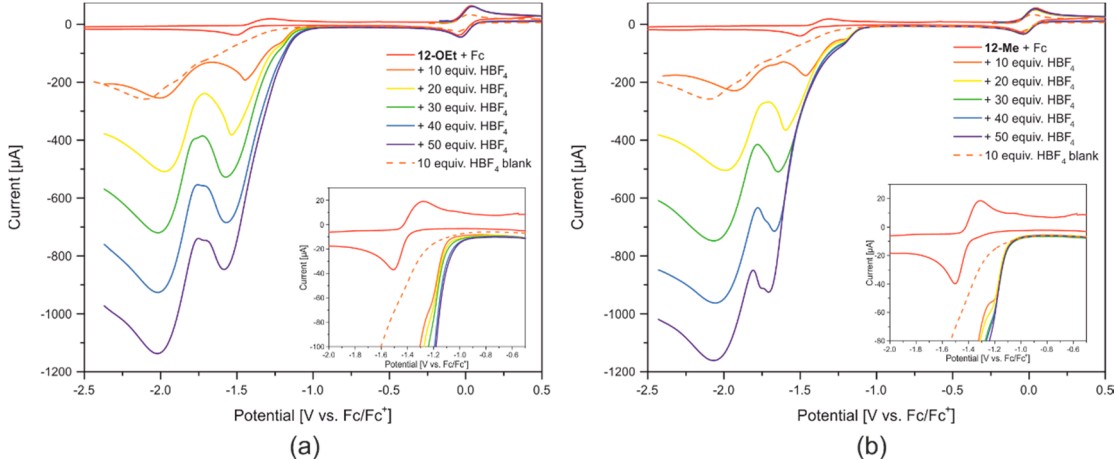

**Figure 11.** (**a**) CVs of **12-OEt**; (**b**) CVs of **12-Me**. Both figures show CVs in presence of 0 to 50 equivalents of HBF$_4$ in H$_2$O. Both measurements were performed using a three-electrode setup utilizing a glassy carbon as working electrode and platinum-and silver wires as counter- and pseudo-reference electrode, respectively. Ferrocene was added as internal reference in all measurements.

Addition of 0.1-2 equivalents of HOAc to **[12-OEt]**$^-$ did not lead to changes within the IR-spectrum (final spectrum see SI Figure S7). Therefore no conclusions regarding the protonation of the reduced phosphinate species **[12-OEt]**$^-$ and the HER mechanism can be drawn from these experiments.

## 3. Conclusions

Three new [FeFe]-hydrogenase model compounds comprising both phosphinate and phosphine oxide bridgeheads were synthesized. Here, bis(mercaptomethyl)derivates in combination with Fe$_3$(CO)$_{12}$ led to substantially better yields than the respective chloride compounds and avoided the synthesis of Fe$_2$S$_2$(CO)$_6$. All three model compounds show identical behavior towards HOAc, undergoing an EC mechanism for catalytic proton reduction. The use of a stronger acid, such es HBF$_4$ in aqueous media or HOTf, led to a change in reactivity. All compounds undergo a CE mechanism, starting with protonation followed by reduction to Fe$^I$Fe$^0$ and the catalytic proton reduction. Due to the preceding protonation, the reduction potential of the [FeFe]-hydrogenase model compounds shift approx. 200 mV towards a more positive potential, showing the decreased electron density at the iron centers. The lowering of the electron density was further approved by IR spectroscopy, which shows a shift of the CO band maxima of 7 cm$^{-1}$ towards higher wavenumbers. Our DFT analysis revealed best agreement between simulated and obtained IR spectra in the case of P=O protonation, while μS-H$^+$, Fe-H$^+$ and Fe-μH$^+$-Fe protonation modes led to stronger CO band shifts or a different band pattern compared to our results. All herein synthesized compounds provide the opportunity to be reduced

to phosphines, which should afford [FeFe]-hydrogenase mimics with $pK_a$ values closer to the native azadithiolate bridge and therefore models with improved catalytic activity that may also fit into the binding pocket of apo-[FeFe]-hydrogenases.

## 4. Materials and Methods

**IR spectroscopy.** Standard IR-measurements were performed with a Bruker Tensor 27 equipped with a Miracle™ single reflection ATR unit (PIKE Technologies) at room temperature. Complexes **12-R** were dissolved and dropped on the Ge crystal. The measurement was started after complete evaporation of the solvent to yield a thin complex film on the ATR crystal.

**Electrochemistry.** Electrochemical testing of the complexes was conducted in acetonitrile using a standard three-electrode setup using a PalmSens3 potentiostat. A glassy carbon (GC) electrode was used as working electrode. An Ag- and a Pt wire were used as quasi-reference and counter electrode, respectively. If not otherwise stated, 0.1 M TBAPF$_6$ was used as the electrolyte and the measured potential was referenced to the ferrocene/ferrocenium couple (Fc/Fc$^+$). Cyclic voltammetry measurements were then performed at a scan rate of 100 mV/s.

**Spectroelectrochemistry**. Spectroelectrochemical measurements were performed with a Bruker Tensor 27, equipped with a Miracle™ single reflection ATR unit (PIKE Technologies) to which a SP-02 cell (Spectroelectrochemistry Partners) was mounted. Al compounds were measured in a concentration of 20 mM. 1 M TBAPF$_6$ was used as the electrolyte. A glassy carbon (GC) electrode was used as working electrode. An Ag- and a Pt wire were used as quasi-reference and counter electrode, respectively. For IR spectra, 128 scans were accumulated to one spectrum at room temperature, while the potential was adjusted using a PalmSens3 potentiostat.

**Mössbauer Spectroscopy**. Zero-field Mössbauer spectra were recorded at 80 K by using a constant acceleration spectrometer equipped with a temperature controller maintaining temperatures within ±0.1 K and a 57Co radiation source in a Rh matrix. Isomer shifts are referred to $\alpha$-Fe metal at room temperature. Data were fit with a sum of Lorentzian quadrupole doublets by using a least-square routine with WMOSS program.

**X-ray Data Collection and Structure Solution Refinement**. Single crystals suitable for X-ray structure analysis were coated with Paratone N oil, mounted on a fiber loop, and placed in a cold stream in the diffractometer. For **12-OEt** Oxford XCalibur diffractometer performing $\varphi$ and $\omega$ scans at 170(2) K was used. Diffraction intensities were measured using graphite-monochromatic Mo K$\alpha$ radiation ($\lambda$ = 0.71073 Å). Data collection, indexing, initial cell refinements, frame integration, final cell refinements, and absorption corrections were accomplished with the program CrysAlisPro.[5] Space groups were assigned by analysis of the metric symmetry and systematic absence (determined by XPREP of WinGX[6]) and were further checked by PLATON[7, 8] for additional symmetry. Structures were solved by direct methods and refined against all data in the reported $2\theta$ ranges by full-matrix least square on $F^2$ with the SHELXL[9] program suite using the shelXle interface.[10, 11] Crystallographic data as well as refinement parameters are presented in Table S1.

*Syntheses*

**(HOCH$_2$)$_2$P(O)OH (1)**. Phosphinic acid (33 g, 0.5 mol), paraformaldehyde (30 g, 1 mol) and 38 mL HCl (37%) were combined in a 250 mL single necked flask and stirred at 40–45 °C for approx. 6 h, until the mixture turns into a colorless solution, which was stirred at reflux conditions for additional 30 h. Excess water was removed using a rotary evaporator and additional azeotropic distillation with toluene, yielding 120 g (0.5 mol, 100%) a slightly yellow viscous oil. **¹H NMR** (250 MHz, D$_2$O, ppm): 3.85 (d, $^2J_{PH}$ = 5.2 Hz, 4H, -CH$_2$OH). **³¹P NMR (101 MHz, D$_2$O, ppm):** 45.4 (s). **ESI-MS** (QP, positive): 126.9 [M+H]$^+$. **Refractive index:** $n_D^{21}$ = 1.502.

**(ClCH$_2$)$_2$P(O)Cl (2)**. **1** (95 g, 0.75 mol) was added very slowly (15 g/h) to 100 mL refluxing SOCl$_2$, resulting in extensive formation of SO$_2$ and HCl. After gas evolution ceased, the reaction mixture was distilled to yield excess SOCl$_2$ at 90 °C, 1013 mbar and **2** at 150 °C, 0.2 mbar with 88% yield and 91%

purity according to $^{31}$P NMR. **$^1$H NMR** (250 MHz, CDCl$_3$, ppm): 4.06 (m, 4H, -**CH$_2$Cl**). **$^{13}$C NMR** (63 MHz, CDCl$_3$, ppm): 37.14 (d, $^1$J$_{PC}$ = 90 Hz). **$^{31}$P NMR** (101 MHz, CDCl$_3$, ppm): 51.0 (s). **ESI-MS** (QP, positive): 163.0 [M+H$_2$O −Cl$^-$]$^+$ (from hydrolyzation). **Refractive index:** $n_D^{21}$ = 1.519

**(ClCH$_2$)$_2$P(O)OEt (3-Et).** **2** (5g, 27.8 mmol) was dissolved in 65 mL dry THF and cooled to 0 °C. To that solution 4.62 mL (33.3 mmol, 1.2 equivalents) N(Et)$_3$ was added. To this now orange solution, 1.94 mL (33.3 mmol, 1.2 equivalents) ethanol was added drop wise and stirred for 4 h at room temperature. A white precipitate was filtered off and the filtrate concentrated to a red oil that was purified via distillation (head temperature 100-105 °C, 0.86 mbar) and follow up bulb-to-bulb distillation (160 °C, 1.4 mbar) to yield 3.2 g (60%) of a colorless oil. **$^1$H NMR** (250 MHz, CDCl$_3$, ppm): 1.26 (td, $^3$J$_{HH}$ = 7.1 Hz, $^4$J$_{PH}$ = 2.8 Hz, 3H, OCH$_2$**CH$_3$**), 3.62 (dd, $^2$J$_{HH}$ = 2.2 Hz, $^2$J$_{PH}$ = 8.5 Hz, 4H, -**CH$_2$**P), 4.11 (m, 2H, O**CH$_2$**CH$_3$) **$^{13}$C NMR** (63 MHz, CDCl$_3$, ppm): 16.3 (d, $^3$J$_{PC}$ = 5,7 Hz, OCH$_2$**CH$_3$**), 32.6 (d, $^1$J$_{PC}$ = 106 Hz, -**CH$_2$**-P), 62.8 (d, $^2$J$_{PC}$ = 7 Hz, O**CH$_2$**CH$_3$). **$^{31}$P NMR** (101 MHz, CDCl$_3$, ppm): 39.4 (s). **Refractive index:** $n_D^{21}$ = 1.502

**(ClCH$_2$)$_2$P(O)OMe (3-Me).** **2** (10 g, 56 mmol) was dissolved in 100 mL dry THF and cooled to 0 °C. To that solution, 9.25 mL (66.7 mmol, 1.2 equivalents) N(Et)$_3$ was added. To this now orange solution, 2.14 g (66.7 mmol, 1.2 equivalents) methanol was added and stirred for 4 h at room temperature. A white precipitate was filtered off and the filtrate concentrated to an orange to red oil that was purified via bulb-to-bulb distillation (130 °C, 0.9 mbar) to yield 8.7 g (88%) of a colorless oil. **$^1$H NMR** (400 MHz, CDCl$_3$, ppm): 3.69 (d, $^2$J$_{PH}$ = 8.0 Hz, 4H, -**CH$_2$Cl**), 3.82 (d, $^3$J$_{PH}$ = 12 Hz, 3H, -**OCH$_3$**). **$^{13}$C NMR** (100 MHz, CDCl$_3$, ppm): 32.3 (d, $^2$J$_{PC}$ = 107 Hz, -**OCH$_3$**), 53.0 (d, $^1$J$_{PC}$ = 7.2 Hz, -**CH$_2$Cl**). **$^{31}$P NMR** (162 MHz, CDCl$_3$, ppm): 41 (s).

**(ClCH$_2$)$_2$P(O)OPh (3-Ph).** **2** (10 g, 56 mmol) was dissolved in 100 mL dry THF and cooled to 0 °C. To that solution 9.25 mL (66.7 mmol, 1.2 equivalents) N(Et)$_3$ was added. To this now orange solution, 6.26 g (66.7 mmol, 1.2 equivalents) phenol was added and stirred for 4 h at room temperature. A white precipitate was filtered off and the filtrate concentrated to a red oil that was purified via bulb-to-bulb distillation (130 °C, 0.9 mbar) to yield 10.8 g (80%) of a slightly yellowish oil. **$^1$H NMR** (400 MHz, CDCl$_3$, ppm): 3.78 (d, $^2$J$_{PH}$ = 8.5 Hz, 4H, -**CH$_2$Cl**), 6.83 (m, 1H, arom.), 7.1–7.5 (m, 4H, arom.). **$^{13}$C NMR** (100 MHz, CDCl$_3$, ppm): 32.6 (d, $^1$J$_{PC}$ = 106 Hz, -**CH$_2$Cl**), 115.5 (s, arom.), 120.8 (d, J$_{PC}$ = 4.3 Hz, arom.), 126.1 (d, J$_{PC}$ = 1.6 Hz, arom.), 130.3 ppm (d, J$_{PC}$ = 1.4 Hz, arom.) **$^{31}$P NMR** (162 MHz, CDCl$_3$, ppm): 39.4 (s). **ESI-MS** (QP, positive): 238.8 [M+H]$^+$.

**(ClCH$_2$)$_2$P(O)CH$_3$ (4).** Methylmagnesium bromide (27 mL, 83 mmol) was dissolved in 200 mL dry diethyl ether and cooled to 0 °C. The solution was mechanically stirred while **2** (10 g, 66 mmol), dissolved in 50 mL dry ethyl ether, was slowly added via dropping funnel. After addition, the mixture was stirred at 50 °C for 4 h and quenched with a saturated solution of sodium carbonate, resulting in strong heat development. The obtained solid was filtered off and washed with chloroform. The filtrate was extracted with a total amount of 400 mL chloroform. The organic phases were combined and all solvents removed by a rotavapor and a high vacuum oil pump, resulting in 4 g (25 mmol, 38%) of a colorless oil that crystallizes after complete loss of solvent. **$^1$H NMR** (250 MHz, CDCl$_3$, ppm): 1.76 (d, $^2$J$_{PH}$ = 13.2 Hz, 3H, -**CH$_3$**), 3.76 (m, 4H, -**CH$_2$Cl**). **$^{13}$C NMR** (100 MHz, CDCl$_3$, ppm): 10.0 (d, $^1$J$_{PC}$ = 73 Hz, -**CH$_3$**), 34.7 (d, $^1$J$_{PC}$ = 71 Hz, -**CH$_2$Cl**). **$^{31}$P NMR** (102 MHz, CDCl$_3$, ppm): 42.6 (s). **ESI-MS** (QP, positive): 161.0 [M]$^+$. **XRD**: Figure S3

**(ClCH$_2$)$_2$P(O)OH (5).** **2** (5 g, 27.8 mmol) was slowly added to 5 mL water. After addition, the solution was filtered and excess water removed using a rotary evaporator, yielding in 4.1 g (91%, 98% purity according to $^{31}$P NMR) of **5** as white solid. **$^1$H NMR** (400 MHz, D$_2$O, ppm): 3.18 (d, $^2$J$_{PH}$ = 8 Hz, 4H, -**CH$_2$Cl**). **$^{13}$C NMR** (100 MHz, D$_2$O, ppm): 34.2 (d, $^1$J$_{PC}$ = 103 Hz). **$^{31}$P NMR** (162 MHz, D$_2$O, ppm): 33.0 (s).

**(HSCH$_2$)$_2$P(O)OH (6).** **5** (4 g, 24.7 mmol) and thiourea (3.76 g, 50 mmol) were mixed in 15 mL n-Butanol and stirred at 115 °C for 6 h and at room temperature overnight. A white precipitate was filtered off and washed with EtOH and dried under reduced pressure. An amount of 20 mL 6 M KOH was added, and the mixture stirred for 1 h at 100 °C. The solution was then acidified with 10 mL 12 M

HCl and washed with DCM. The aqueous phase was narrowed to yield a white sticky solid that was dispersed in EtOH and filtered. The solvent was the ethanolic solution was removed to yield a slightly milky oil. **$^1$H NMR** (400 MHz, D$_2$O, ppm):1.90 (2H, **-SH**), 2.90 (4H, **-CH$_2$SH**), 4.51 (1H, **POH**). **$^{31}$P NMR** (162 MHz, D$_2$O, ppm): 22.9(s).

**(AcSCH$_2$)$_2$P(O)OEt (7-Et)**. **3-Et** (5.7 g, 30 mmol) was dissolved in 60 mL dry DMF and potassium thioacetate (8.6 g, 75 mmol) was added. The red turning mixture was stirred at ambient temperature overnight. The solvent was removed, the residue dissolved in DCM and washed with water. Finally, the organic phase was narrowed to yield 6.4 g (23.3 mmol, 78%) of a dark red oil. **$^1$H NMR** (400 MHz, CDCl$_3$, ppm): 4.12 (dq, 2H, -OCH$_2$-), 3.26 (dd. 4H, -PCH$_2$-) 2.39 (s, 6H, -SC(O)CH$_3$), 1.30 (t, 3H, -OCH$_2$CH$_3$). **$^{13}$C NMR** (100 MHz, CDCl$_3$, ppm): 193 (s, SC(O)-), 62.4 (d, $^2$J$_{PC}$ = 7 Hz, -OCH$_3$-), 30.2 (s, -SC(O)CH$_3$), 24.1 (d, $^1$J$_{PC}$ = 99 Hz, -PCH$_2$-), 16.6 (d, $^3$J$_{PC}$ = 5.8 Hz, -OCH$_2$CH$_3$). **$^{31}$P NMR** (162 MHz, CDCl$_3$, ppm): 43.5 (s).

**(AcSCH$_2$)$_2$P(O)OPh (7-Ph)**. **3-Ph** (7.2 g, 30 mmol) was dissolved in 60 mL dry DMF and potassium thioacetate (8.6 g, 75 mmol) was added. The red turning mixture was stirred at ambient temperature overnight. The solvent was removed, the residue dissolved in DCM and washed with water. Finally, the organic phase was narrowed to yield 8.2 g (25.8 mmol, 86%) of a dark red oil. **$^1$H NMR** (400 MHz, CDCl$_3$, ppm): 7.39-7.20 (m, 5H, arom.), 3.4 (dd. 4H, -PCH$_2$-) 2.41 (s, 6H, -SC(O)CH$_3$). **$^{13}$C NMR** (100 MHz, CDCl$_3$, ppm): 192.7 (s, **S**C(O)-), 130.1 (d, $^2$J$_{PC}$ = 1.2 Hz, arom.), 125.6 (d, $^3$J$_{PC}$ = 1.6 Hz, arom.), 120.7 (d, $^4$J$_{PC}$ = 4.5 Hz, arom.), 115.4 (d, $^5$J$_{PC}$ = 1.4 Hz, arom.), 30.2 (s, -SC(O)**CH$_3$**), 24.2 (d, $^1$J$_{PC}$ = 99 Hz, -PCH$_2$-) **$^{31}$P NMR** (162 MHz, CDCl$_3$, ppm): 43.3 (s).

**(AcSCH$_2$)$_2$P(O)Me (9)**. **4** (5 g, 31 mmol) was dissolved in 60 mL dry DMF and potassium thioacetate (9 g, 78 mmol) was added. The red turning mixture was stirred at ambient temperature overnight. The solvent was removed, the residue dissolved in DCM and washed with water. Finally, the organic phase was evaporated to yield 5.4 g (22.4 mmol, 72%) of a red solid. **$^1$H NMR** (400 MHz, CDCl$_3$, ppm): 1.52 (d, 3H, $^2$J$_{PH}$ = 12.9 Hz, -PCH$_3$), 2.41 (s, 6H, SC(O)CH$_3$), 3.28 (d, 4H $^2$J$_{PH}$ = 9 Hz, -PCH$_2$-). **$^{13}$C NMR** (100 MHz, DMSO, ppm): 13.2 (d, $^1$J$_{PC}$ = 71.6 Hz, -PCH$_3$), 25.6 (d, $^1$J$_{PC}$ = 66.1 Hz, -PCH$_2$-), 30.0 (s, -SC(O)CH$_3$), 193.4 (d, $^3$J$_{PC}$ = 2.8 Hz, SC(O)-). **$^{31}$P NMR** (162 MHz, CDCl$_3$, ppm): 43.4 (s). **ESI-MS** (QP, positive): 262.7 [M+Na]$^+$, 240.8 [M+H]$^+$, 198.9 [M –(CH$_3$CO) +2H]$^+$, 157.3 [M -2(CH$_3$CO) +3H]$^+$.

**(HSCH$_2$)$_2$P(O)OEt (8-Et)**. **7-Et** (2.7 g, 10 mmol) was dissolved in 50 mL MeOH and NaOH (3 M in water, 20 mL) was added. The solution was kept stirring at room temperature for 3 h, neutralized with 10 mL 6 M HCl and finally extracted with DCM. The organic phase was evaporated to yield a dark red oil that was purified by bulb-to-bulb distillation to obtain a yellow oil.

*Note: Besides ethyl bis(mercaptomethyl) phosphinate, this procedure gives 4-ethoxy-1,2,4-dithiaphospholane 4-oxide, or a polymer by forming inter- instead of intramolecular disulfide bridges, as well. Those oxidized products can be specifically synthesized by the following procedure:*

Ethyl bis((acetyl)methyl) phosphinate (1.5 g, 5.6 mmol) was dissolved in 10 mL degassed, dry THF. Furthermore, sodium (255 mg, 11.2 mmol) was dissolved in 5 mL MeOH, forming NaOMe, and added dropwise to the THF solution. After the addition is complete, 100 mL THF was added for dilution before 1.41 g (6 mmol) I$_2$, dissolved in 50 mL THF was added very slowly. This mixture was stirred for 3 d at RT before the solvent was removed yielding a greasy residue that was extracted with DCM. The solvent of the organic phase was removed to afford a dark red highly viscous oil that did not require further purification. **$^1$H NMR** (400 MHz, CDCl$_3$, ppm): <u>Dithiol</u>: 1.17 (t, $^3$J$_{HH}$ = 7 Hz, 3H, -OCH$_2$CH$_3$), 1.77 (s, 2H. –SH), 2.68 (dd, $^2$J$_{HH}$ = 5.3 Hz, $^2$J$_{PH}$ = 10.4 Hz, 4H, -CH$_2$SH), 3.99 (dq, $^3$J$_{HH}$ = 7 Hz, $^3$J$_{PH}$ = xx Hz, 2H, -OCH$_2$CH$_3$). <u>Dithiaphospholan</u>: 1.41 17 (t, $^3$J$_{HH}$ = 7 Hz, 3H, -OCH$_2$CH$_3$), 3.06 (dd, $^2$J$_{HH}$ = 6.6 Hz, $^2$J$_{PH}$ = 8.5 Hz, 4H, -CH$_2$S), 4.22 (dq, $^3$J$_{HH}$ = 7 Hz, $^3$J$_{PH}$ = 7.9 Hz, 2H, -OCH$_2$CH$_3$). **$^{13}$C NMR** (100 MHz, CDCl$_3$, ppm): <u>Dithiol</u>: 17.5 (d, $^3$J$_{PC}$ = 98 Hz, -OCH$_2$CH$_3$), 29.8 (d, $^1$J$_{PC}$ = 93 Hz, -CH$_2$SH), 62.0 (d, $^2$J$_{PC}$ = 7 Hz, -OCH$_2$CH$_3$). <u>Dithiaphospholan</u>: 18.9 (d, $^3$J$_{PC}$ = 97 Hz, -OCH$_2$CH$_3$), 33.7 (d, $^1$J$_{PC}$ = 93 Hz, -CH$_2$S), 62.4 (dq, $^2$J$_{PC}$ = 7 Hz, -OCH$_2$CH$_3$). **$^{31}$P** NMR (162 MHz, CDCl$_3$, ppm): <u>Dithiol</u>:

48 (s); Dithiaphospholane: 80 (s). **ESI-MS** (QP, positive): 186.6 $[M+H]^+$, 159.0 $[M+H_2O -OEt^-]^+$ (from hydrolyzation).

**$(HSCH_2)_2P(O)OPh$ (8-Ph).** **7-Ph** (3.5 g, 11 mmol) was dissolved in 50 mL MeOH and NaOH (3 M in water, 22 mL) was added. The solution was kept stirring at room temperature for 3 h, neutralized with 11 mL 6 M HCl and finally extracted with DCM. The organic phase was evaporated to yield a dark red oil that was purified by bulb-to-bulb distillation to obtain a yellow oil.

*Note: Besides phenyl bis(mercaptomethyl) phosphinate, this procedure gives 4-phenoxy-1,2,4-dithiaphospholane 4-oxide, or a polymer by forming inter- instead of intramolecular disulfide bridges, as well. Here, only $^{31}$P-NMR spectroscopy was able to distinguish between both products*

**$^1$H NMR** (400 MHz, CDCl$_3$, ppm): 1.76 (dt, 2H, –SH), 2.78 (m, 8H, -**CH$_2$**SH and -**CH$_2$**S-), 6.78-7.12 (m, 5H, arom., overlapped with toluene), **$^{13}$C NMR** (100 MHz, CDCl$_3$, ppm): 17.0 (d, $^3J_{PC}$ = 100 Hz, -**CH$_2$**SH), 115.5 (s, arom.), 120.6 (s, arom.), 129.8 (s, arom.), 155.8 (s, arom.). **$^{31}$P NMR** (162 MHz, CDCl$_3$, ppm): Dithiol: 52 (s); Dithiaphospholane: 84 (s). **ESI-MS** (QP, positive): 238.8 $[M+H]^+$.

**$(HSCH_2)_2P(O)CH_3$ (10).** **9** (1 g, 4.17 mmol) was dissolved in 30 mL MeOH and NaOH (3 M in water, 8.34 mL) was added. The solution was kept stirring at room temperature for 0.5 h, neutralized with 40 mL 1 M HCl and finally extracted with chloroform. The organic phase was evaporated to yield a dark red oil that was purified by bulb-to-bulb distillation to obtain a smelly yellow oil.

*Note: Besides Bis(mercatptomethyl)methyl phosphine oxide, this procedure gives 4-methyl-1,2,4-dithiaphospholane 4-oxide, or a polymer by forming inter- instead of intramolecular disulfide bridges, as well.*

**$^1$H NMR** (400 MHz, CDCl$_3$, ppm): Dithiol: 1.68 (d, $^2J_{PH}$ = 12.6 Hz, 3H, -**CH$_3$**), 1.94 (td, $^3J_{HH}$ = 6.6 Hz, $^3J_{PH}$ = 8.5 Hz, 2H. –SH), 2.68 (dd, $^2J_{HH}$ = 5.3 Hz, $^2J_{PH}$ = 10.4 Hz, 4H, -**CH$_2$**SH). **$^{13}$C NMR** (100 MHz, CDCl$_3$, ppm): Dithiol: 17.5 (d, $^3J_{PC}$ = 98 Hz, -O**CH$_2$CH$_3$**), 29.8 (d, $^1J_{PC}$ = 93 Hz, -**CH$_2$**SH), 62.0 (d, $^2J_{PC}$ = 7 Hz, -O**CH$_2$**CH$_3$). **$^{31}$P NMR** (162 MHz, CDCl$_3$, ppm): Dithiol: 47 (s). Dithiaphospholane: 79 (s). **ESI-MS** (QP, positive): 156.9 $[M]^+$.

**Syntheses of model compounds 12-R. Method A.** Fe$_2$S$_2$(CO)$_6$ (100 mg, 0.29 mmol) was dissolved in 5 mL dry THF in a 50 mL single necked Schlenk flask, which was sealed with a rubber septum and equipped with a bubbler. The reaction vessel was cooled to −78 °C under a constant slight Ar flow to avoid pressure differences, before 2.2 equivalents of LiBHEt$_3$ (1 M in THF) was added via syringe drop wise. This addition resulted in a color change from orange over purple to dark green, indicating the formation of Li$_2$[Fe$_2$S$_2$(CO)$_6$]. After 15 min, **3-Et** (75 mg, 0.4 mmol) and NEt$_3$ (83 μL, 0.6 mmol) were added consecutively via syringe and the reaction mixture allowed to warm to room temperature overnight. The solvent was completely removed with a liquid N$_2$ trap, before the residual red solid was dissolved in a mixture from PE and EtOAc (1:1), filtered and finally purified via column chromatography with PE/EtOAc (1:1) as mobile phase and SiO$_2$ as stationary phase.

**Method B.** Fe$_3$(CO)$_{12}$ (500 mg, 1 mmol) was dissolved in 20 mL dry THF in a 50 mL single necked Schlenk flask, which was sealed with a rubber septum and equipped with a bubbler. To that green solution, 1.3 equivalents of dithiol were added via syringe. The reaction mixture was then kept stirring at room- or slightly elevated temperature (<40 °C) until it became intense red and no starting material was left according to TLC (elution with PE, approx. 2–3 h). The solvent was removed by a cold trap and the residual solid dissolved in the eluent for following chromatographic purification.

**12-OEt.** Elution with PE/EtOAc (1:1). **IR** ($\upsilon$CO, cm$^{-1}$): 2077, 2038, 1999, 1975. **EA:** calcd. for C$_{10}$H$_9$Fe$_2$O$_8$PS$_2$: C, 25.89; H, 1.96; Fe, 24.07; O, 27.59; P, 6.68; S, 13.82. found: C, 25.48; H, 2.22; S, 13.65

**12-OMe.** Elution with DCM/MeOH (50:1). **IR:** ($\upsilon$CO, cm$^{-1}$): 2077, 2034, 1985.

**12-Me.** Elution with DCM/MeOH (25:1). **IR:** ($\upsilon$CO, cm$^{-1}$): 2077, 2041, 1996. **EA:** calcd. for C$_9$H$_7$Fe$_2$O$_7$PS$_2$: C, 24.91; H, 1.63; Fe, 25.74; O, 25.81; P, 7.14; S, 14.78. found: C, 24.83; H, 1.75; S, 14.65

**Supplementary Materials:** The following are available online at http://www.mdpi.com/2073-4344/10/5/522/s1, Computational Methodology, Figure S1: Complete asymmetric unit of compound 12-OMe, Figure S2: Molecular structure of 12-Me, Figure S1: IR spectra of compounds 12-R in presence of 0–50 equivalents of HBF$_4$ (33% in H$_2$O) in acetonitrile at room temperature, Figure S4: DFT derived IR spectra of 12-Me in its neutral and protonated forms, Figure S5: Cyclic voltammograms of P(O)OMe (12-OMe) in absence and presence of HBF$_4$ (33% in H$_2$O), Figure S6: IR-difference spectrum of the reduction of 12-OEt, Figure S7: Comparison of IR-Spectra of 12-OEt, [12-OEt]$^-$ and [12-OEt]$^-$ after addition of 2 equivalents HOAc, Figures S8-S52: NMR Spectra of compounds 1–10. Table S1: Crystallographic data and refinement parameters of compounds 12-R.

**Author Contributions:** Conceptualization, F.W. and U.-P.A.; methodology, F.W., E.B.B; syntheses, F.W., calculations, E.B.B; writing—original draft preparation, F.W.; writing—review and editing, F.W., E.B.B., U.-P.A., M.R.; supervision, U.-P.A., M.R.; funding acquisition, U.P.A. All authors have read and agreed to the published version of the manuscript.

**Funding:** F.W. thanks the Studienstiftung des Deutschen Volkes for a PhD fellowship. U.-P.A. acknowledges funding by the Fonds der Chemischen Industrie (Liebig Grant), the DFG (Emmy Noether Grant AP242/2-1, AP242/12-1), and the Fraunhofer Internal Programs (Grant ATTRACT 097-602175). EBB and MR thank the DFG for funding through Emmy-Noether project RO5688/1-1.

**Conflicts of Interest:** The authors declare no conflict of interest.

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
