# Peer review of "New Phosphorous-Based [FeFe]-Hydrogenase Models"

_catalysts, doi:10.3390/catal10050522_

Round 1

Reviewer 1 Report

The manuscript reports the synthesis and characterization of models of the diiron cofactor of FeFe-hydrogenases’ active site models in which the secondary amine of the bridging ligand is replaced by phosphinates or phosphine oxides. The results are clearly presented and solid enough to support the conclusions of the work. However, the need of including the last part of the manuscript (subheading 2.5 of results section)about the synthesis of models with reduced phosphorus groups is not justified because those results are incomplete (the assembly of the diiron cofactor including these groups did not work out).

Minor revision is required for correcting the following points:

  • Page 1, line 31: It should be Figure 1A instead of Figure 2A
  • Caption of Figure 1: The colour code for atoms of the structure should be included.
  • Page 2, line 54: It should be Figure 2A instead of Figure 1A
  • Caption of Figure 4: It should be trimethylphosphine.
  • Scheme 1: The formula for paraformaldehyde in reaction A is incomplete. In reaction B it should say heat instead of neat
  • Description of Scheme 1 in the text: Part of the information is redundant, as it is also indicated in the Materials and Methods section.
  • There are two Figures 7.
  • The resolution of the Mossbauer spectra lines is too low.
  • The size and resolution of the numbers and names in the infrared graphs is too small.
  • Page 11, line 316: The electrocatalytic reduction wave is at -2.2 V not -2.4 V.
  • Page 11, line 328: It should be Figure S5.
  • Page 12, line 366: Albracht and co-workers were the first to report the "self-cannibalization” effect in FeFe-hydrogenases.
  • Page 13, line 405: It should be Scheme 3.
  • Page 13, line 407: It should be indicated that the Method B is that of Scheme 2.

Author Response

The manuscript reports the synthesis and characterization of models of the diiron cofactor of FeFe-hydrogenases’ active site models in which the secondary amine of the bridging ligand is replaced by phosphinates or phosphine oxides. The results are clearly presented and solid enough to support the conclusions of the work. However, the need of including the last part of the manuscript (subheading 2.5 of results section)about the synthesis of models with reduced phosphorus groups is not justified because those results are incomplete (the assembly of the diiron cofactor including these groups did not work out).

We thank this referee for his statement and contribution to improve our manuscript. In accordance with his remark on section 2.5, we removed this paragraph from the manuscript as well as the referenced Supporting Information.

Page 1, line 31: It should be Figure 1A instead of Figure 2A

Page 2, line 54: It should be Figure 2A instead of Figure 1A

There are two Figures 7.

Page 11, line 328: It should be Figure S5.

Page 13, line 405: It should be Scheme 3.

We thank the referee for noting the disorder of the figures, we adjusted all figures and schemes in the right order.

Caption of Figure 1: The colour code for atoms of the structure should be included.

We added the color code in the caption of Figure 1 and in the caption of Figure 5 as well.

Caption of Figure 4: It should be trimethylphosphine.

We corrected that typo

Scheme 1: The formula for paraformaldehyde in reaction A is incomplete. In reaction B it should say heat instead of neat

We corrected the formula for paraformaldehyde. In reaction B “neat” was intended. To avoid misleading wording, we deleted this phrase. We thank the referee for this hint.

Description of Scheme 1 in the text: Part of the information is redundant, as it is also indicated in the Materials and Methods section.

The referee is right. We deleted the redundant information.

The resolution of the Mossbauer spectra lines is too low.

The size and resolution of the numbers and names in the infrared graphs is too small.

We adjusted the figures according to the suggestions of this referee.

Page 11, line 316: The electrocatalytic reduction wave is at -2.2 V not -2.4 V.

We corrected this mistake. The new sentence reads as follows: After addition of 10 equivalents acetic acid, a catalytic reduction wave can be observed at −2.2 V vs Fc/Fc+ after reducing the H2ase mimic to its FeIFe0 state that increases from 10 to 20 equivalents and is at its maximum to 30 equivalents

Page 12, line 366: Albracht and co-workers were the first to report the "self-cannibalization” effect in FeFe-hydrogenases.

We thank this referee for this note. We changed the reference for this paragraph from Silakov et al. to Albracht et al.

Page 13, line 407: It should be indicated that the Method B is that of Scheme 2.

We thank the referee for this hint. However, since we deleted to whole paragraph, this remark is negligible.

Reviewer 2 Report

The authors presented a work of great impact and soundness entitled: ‘’New Phosphorous-based [FeFe]-Hydrogenase Models’'. Whole work result to be well defended and well designed.  However, additional work is needed before final publication, see below.

Line 31 (Figure 2A).[1–4] In accordance with journal guidelines the current style is not correct. Whole manuscript must be revisited.

Line 41 Figure 1: A) Same problem as above. Whole manuscript must be revisited.

Lines 72-73 are empty.

Line 85 (Figure 3, B) Change by following the journal guidelines. Whole manuscript must be revisited.

Line 106 (D, ref [17]) and (E, ref [17])

Line 662 References must to follow Catalyst style.

There are some typos in the manuscript.

Author Response

The authors presented a work of great impact and soundness entitled: ‘’New Phosphorous-based [FeFe]-Hydrogenase Models’'. Whole work result to be well defended and well designed.  However, additional work is needed before final publication, see below.

We thank this referee for his comments to improve our manuscript.

Line 31 (Figure 2A).[1–4] In accordance with journal guidelines the current style is not correct. Whole manuscript must be revisited.

Line 85 (Figure 3, B) Change by following the journal guidelines. Whole manuscript must be revisited.

Line 41 Figure 1: A) Same problem as above. Whole manuscript must be revisited.

We thank this referee for checking the style. We adjusted all figures and captions according to the guidelines. We also changed the text in order to stick to the style.

Lines 72-73 are empty.

We deleted one empty line, however there is still one line ahead of table 1 for better readability.

Line 106 (D, ref [17]) and (E, ref [17])

We changed the caption of these figures that it reads as follows: “(D)[17], trimethylphosphine (E)[17]” according to the reference style guide

Line 662 References must to follow Catalyst style.

We checked the style recommendations and updated the Reference list using the suggested mdpi style with Zotero.

There are some typos in the manuscript.

We thank the referee for taking care of spelling. We carefully read through the document to correct the typos.